# Extending Unsupervised Neural Image Compression With Supervised Multitask Learning

**David Tellez**                DAVID.TELLEZMARTIN@RADBOUDUMC.NL
*Department of Pathology, Radboud University Medical Center, Nijmegen, The Netherlands*

**Diederik Höppener**               D.HOPPENER@ERASMUSMC.NL
**Cornelis Verhoef**                C.VERHOEF@ERASMUSMC.NL
**Dirk Grünhagen**                D.GRUNHAGEN@ERASMUSMC.NL
**Pieter Nierop**                 P.NIEROP@ERASMUSMC.NL
*Department of Surgical Oncology and Gastrointestinal Surgery, Erasmus MC Cancer Institute, Rotterdam, The Netherlands*

**Michal Drozdzal**                MDROZDZAL@FB.COM
*Facebook AI Research*

**Jeroen van der Laak**          JEROEN.VANDERLAAK@RADBOUDUMC.NL
**Francesco Ciompi**           FRANCESCO.CIOMPI@RADBOUDUMC.NL
*Department of Pathology, Radboud University Medical Center, Nijmegen, The Netherlands*

**Editors:** Accepted for MIDL 2020

## Abstract

We focus on the problem of training convolutional neural networks on gigapixel histopathology images to predict image-level targets. For this purpose, we extend Neural Image Compression (NIC), an image compression framework that reduces the dimensionality of these images using an encoder network trained unsupervisedly. We propose to train this encoder using supervised multitask learning (MTL) instead. We applied the proposed MTL NIC to two histopathology datasets and three tasks. First, we obtained state-of-the-art results in the Tumor Proliferation Assessment Challenge of 2016 (TUPAC16). Second, we successfully classified histopathological growth patterns in images with colorectal liver metastasis (CLM). Third, we predicted patient risk of death by learning directly from overall survival in the same CLM data. Our experimental results suggest that the representations learned by the MTL objective are: (1) highly specific, due to the supervised training signal, and (2) transferable, since the same features perform well across different tasks. Additionally, we trained multiple encoders with different training objectives, e.g. unsupervised and variants of MTL, and observed a positive correlation between the number of tasks in MTL and the system performance on the TUPAC16 dataset.

**Keywords:** Neural image compression, supervised multitask learning, histopathology

## 1. Introduction

Pathologists examine whole-slide images (WSIs) to diagnose a wide variety of diseases and predict patient prognosis. These WSIs are gigapixel images of human tissue sections taken at very high resolution, i.e. subcellular detail. In order to perform WSI classification, pathologists incorporate visual features from the entire WSI at once. This task poses two main challenges to Computer Vision algorithms: first, processing images of gigapixel resolution at once is computationally extremely expensive; and, second, the low signal-to-noise ratio

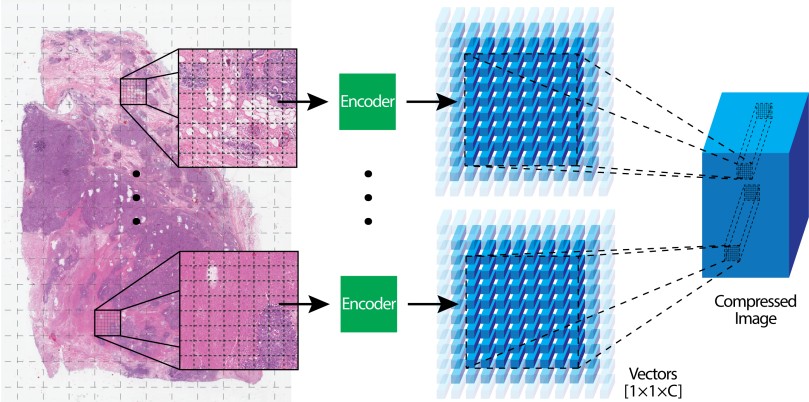

Figure 1: Neural Image Compression. Left: an entire gigapixel whole-slide image is read as a set of high-resolution patches using a uniform grid. Center: each of these patches is compressed into a low-dimensional embedding vector using a neural network, the encoder. Right: the embedding vectors are organized following the same spatial arrangement as in the original whole-slide image.

present in WSIs has been shown to limit the performance of these algorithms (Pawlowski et al., 2019).

Several methods have been proposed to solve WSI classification. In multiple-instance learning, a WSI is decomposed into small high-resolution patches (bag of patches) that are weakly annotated using the image label (Xu et al., 2017; Coudray et al., 2018; Wang et al., 2018; Ilse et al., 2018; Combalia and Vilaplana, 2018; Tomczak et al., 2018; Hou et al., 2016; Quellec et al., 2017). However, these methods cannot exploit the relationship between patches, being unable to observe a more global context of the WSI. Reinforcement learning has been proposed as a solution to increase context (Qaiser and Rajpoot, 2019; Dong et al., 2018; BenTaieb and Hamarneh, 2018). Although these methods can integrate knowledge across patches, they suffer from other limitations, e.g. optimization difficulties and leaving large areas of the WSI unexplored. Moreover, other authors have proposed memory-efficient methodologies that enable convolutional neural networks (CNNs) to be trained with very large images (Pinckaers et al., 2019; Kong et al., 2007). However, CNNs struggle to perform well in tasks with very low signal-to-noise ratio like histopathology image analysis (Pawlowski et al., 2019), requiring vast amount of data samples to work. Unfortunately, histopathological datasets rarely surpass the hundreds or thousands of data points (Veta et al., 2019; Bándi et al., 2019), urging for more sample-efficient methods to perform WSI classification.

Neural Image Compression (NIC) is a recently proposed framework (Tellez et al., 2019) that can drastically reduce the dimensionality of WSIs while retaining semantic information and suppressing noise (see Fig. 1). NIC divides each WSI into a set of high-resolution small patches that are independently compressed into embedding vectors using an *encoder*, i.e., a neural network trained unsupervisedly. Then, these vectors are arranged in a 2D grid following the spatial configuration of the original WSI. The result of this operation is a compressed representation of the entire WSI, where each vector corresponds to a patch in

the WSI. Once all WSIs are compressed using NIC, a classifier, e.g. a CNN, is trained on the compressed WSI representations using the image-level labels as targets. NIC addresses the main challenges of WSI classification by reducing both the size and noise levels of WSIs, while allowing the CNN classifier to exploit global context.

A crucial factor of the NIC method is the encoder model. This neural network is responsible for suppressing low-level pixel noise and spurious correlations, while identifying and extracting high-level discriminative features that could work well in a variety of downstream tasks and, as such, should be transferable across a number of histopathological tasks. To satisfy this condition, the original formulation of NIC suggested to use unsupervised methods to train the encoder, such as: variational autoencoders (VAE) (Kingma and Welling, 2013), contrastive training (Koch et al., 2015; Melekhov et al., 2016; Hyvarinen and Morioka, 2016), and adversarial feature learning (Donahue et al., 2017; Dumoulin et al., 2017). Since networks trained with supervised signals are able to extract more specific feature representations than those using unsupervised loss terms (Tan and Le, 2019), we hypothesize that combining multiple supervised goals during training could lead to superior and more generalizable features than using a single unsupervised task. Therefore, we propose to introduce supervision in the training of the encoder and do so via supervised multitask learning (Caruana, 1997). Although, supervised and unsupervised multitask representation learning have shown promising results on multiple Computer Vision benchmarks (Ruder, 2017; Zhang and Yang, 2017), the usefulness of the learned representations for WSI compression is yet an unexplored research avenue. We propose a method that exploits and combines several supervision signals from four representative tasks in Computational Pathology: mitosis detection in breast, axillary lymph node tumor metastasis detection, prostate epithelium detection, and colorectal cancer tissue type classification.

In this work, we trained image compression using the proposed multitask NIC and evaluated the obtained representations in two histopathology datasets that target image-level labels. First, modeling the speed of tumor growth in invasive breast cancer, included in the Tumor Proliferation Assessment Challenge 2016 (TUPAC16) (Veta et al., 2019). Second, predicting histopathological growth patterns and the overall risk of death in patients with colorectal metastasis in the liver (Galjart et al., 2019).

Our contributions can be summarized as follows:

- We improved NIC by training the encoder with supervised multitask learning. Experimental results suggest that embedding vectors were more discriminative and transferable, and adding more tasks to the multitask framework increased the performance of the method at WSI level.

- We obtained state-of-the-art performance predicting tumor proliferation speed in invasive breast cancer patients from the Tumor Proliferation Assessment Challenge, and classifying histopathological growth patterns in patients with colorectal liver metastasis.

- We successfully predicted patient risk of death by learning directly from overall survival in patients with colorectal liver metastasis, without the need for human intervention.

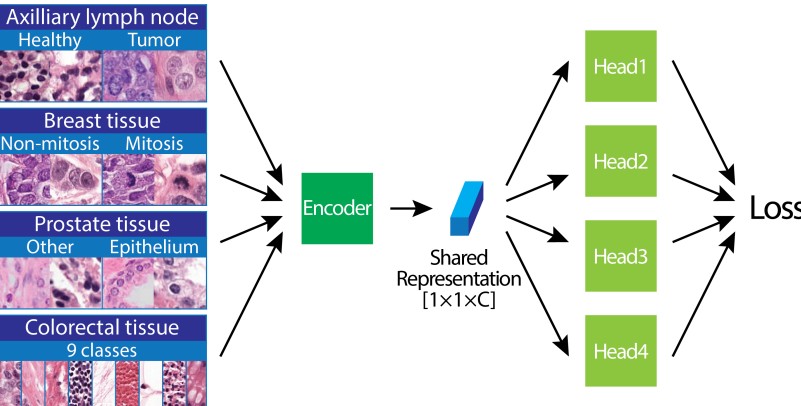

Figure 2: Supervised multitask learning framework. Left: the full model is trained to solve four different tasks simultaneously. Center: the encoder provides a shared embedded representation for the images of all the tasks. Right: the head models perform each of the four classification tasks independently from each other.

## 2. Materials

**Multitask learning dataset**. We selected multicenter data from four representative patch classification tasks in Computational Pathology (see Fig. 2), namely: mitosis detection in breast, axillary lymph node tumor metastasis detection, prostate epithelium detection, and colorectal cancer tissue type classification. A full description of this dataset is available in (Tellez et al., 2019). For each task, we selected 200000 patches of $64 \times 64$ pixels at $0.5\,\mu m$/pixel resolution with patch-level annotations. We distributed the number of patches across classes and medical centers uniformly, so that classes and centers were equally represented in the dataset, and reserved 20% of the samples for validation purposes (randomly selected).

**TUPAC16 dataset**. We used public WSIs from the Tumor Proliferation Assessment Challenge 2016 (TUPAC16) (Veta et al., 2019) to evaluate our method. This cohort consisted of 492 hematoxylin and eosin (H&E) training slides taken from patients with invasive breast cancer from The Cancer Genome Atlas (Weinstein et al., 2013). The organizers of the Challenge provided a label for each patient that served as a proxy for tumor proliferation speed (Nielsen et al., 2010). Additionally, the organizers also provided 321 test slides with no public labels available, that were used to perform a truly independent performance evaluation.

**Colorectal liver metastasis dataset**. This private cohort consisted of 363 patients that underwent colorectal liver metastasis resection at the Erasmus MC Cancer Institute (Rotterdam, the Netherlands) between 2000 and 2015 (Galjart et al., 2019). A total of 1571 H&E stained slides were used in this work. These slides were scanned using a 3DHistech P1000 scanner at a spatial resolution of $0.25\,\mu m$/pixel. Each slide was manually scored for the presence of histopathological growth patterns (HGP) following international guidelines (Van Dam et al., 2017; Höppener et al., 2019). A given slide was considered desmoplastic HGP (dHGP) if this was the only pattern observed, and non-desmoplastic HGP

(non-dHGP) otherwise. The consideration of dHGP is generally associated with better prognosis (longer patient survival). In addition, overall survival was available for these patients, with a mean follow-up period of 3.4 years.

## 3. Methods

**Supervised multitask learning**. Our multitask learning architecture is built from two components. The first component, the encoder, is shared among the four tasks, whereas the second part, the heads, consists of four multilayer perceptrons (MLPs) specialized in solving each task individually. Both the encoder and the four heads are trained to minimize the sum of the classification losses of the four tasks. By doing so, the encoder learns a vector representation that is optimized to produce high classification performance while being highly transferable across different tasks. Fig. 2 provides an overview of the method. The size of the embedded representation $C$ is an hyperparameter of the method, by default set to $C = 128$ following the original implementation of NIC. We trained this model using images from the *multitask learning dataset* only.

The architecture of the encoder consisted of 4 strided convolutional layers with 128 $3 \times 3$ filters, batch normalization, leaky-ReLU activation (LRA), and stride of 2; followed by a linear layer with $C$ units. The head models were composed of a dense layer with 256 units, and LRA, with 10% dropout before and after this layer; and a final dense layer whose number of output units depended on the classification task (9 for the multiclass tissue classification, and 2 for the rest), followed by a softmax.

We trained the encoder and head models simultaneously by minimizing the average categorical cross-entropy across the four tasks. We used stochastic gradient descent with Adam optimization and a mini-batch of 128 samples (32 samples per task dataset), decreasing the learning rate by a factor of 10 starting from $1 \times 10^{-3}$ every time the validation metric plateaued until $1 \times 10^{-5}$. During training, we used heavy morphological and color augmentation (Tellez et al., 2019), increasing the model robustness to unseen data.

**WSI classification**. In order to train a CNN classifier on gigapixel WSIs and image-level labels, we followed the method described in the original NIC publication, with the exception of using the proposed multitask encoder instead of the unsupervised model. A detailed description of the CNN architecture and training details is available in the Appendix A.

**Learning from patient overall survival**. Survival analysis constitutes a regression problem where a model is trained to predict a risk score for each patient that is proportional to their chances of experiencing the event of death. Each patient's WSI is associated with a record composed of two items: a follow-up period and a binary death-event variable. We used WSIs compressed with multitask NIC and overall survival data to train a CNN classifier to predict patient risk of death by maximizing the partial log-likelihood loss (Faraggi and Simon, 1995). Intuitively, by optimizing this objective the CNN classifier learned to assign high risk scores to those patients that died early. See Appendix C for a detailed description of this loss.

Table 1: Predicting tumor proliferation speed in TUPAC16 Spearman corr. and 95% c.i.

| Method | Training set | External test set |
|---|---|---|
| TUPAC16 top-3 (Veta et al., 2019) | - | 0.503 |
| TUPAC16 top-2 (Veta et al., 2019) | - | 0.516 |
| NIC unsupervised (Tellez et al., 2019) | 0.522 | 0.558 [0.5191, 0.5962] |
| Streaming CNNs (Pinckaers et al., 2019) | - | 0.570 |
| TUPAC16 top-1 (Veta et al., 2019) | - | 0.617 |
| NIC multitask (proposed) | 0.620 | **0.632 [0.5966, 0.6641]** |
| TUPAC16 human-assisted (*) (Veta et al., 2019) | - | 0.710 |

(*) Requires the intervention of an expert pathologist

## 4. Experimental results

### 4.1. Training the multitask encoder

For the purpose of compressing WSIs in the TUPAC16 and liver datasets, we first trained a 4-task multitask encoder following the procedure described in the Methods section. We obtained the following validation accuracy scores at patch level: lymph node tumor classification (90.87%), mitosis classification (94.81%), prostate epithelium classification (86.48%), and colorectal 9-class classification (77.49%).

### 4.2. Predicting the speed of tumor proliferation (TUPAC16)

In this experiment, we used the previously trained encoder to compress the WSIs on the TUPAC16 training dataset; and trained four CNN regressors on top of these compressed WSIs using 4-fold cross-validation (3 folds for training, 1 for validation). For the test set, we compressed the WSIs similarly, then averaged the predictions of the four CNNs per sample, and submitted the results to the Challenge organizers for independent evaluation (the labels of the test set are not public). Our proposed method achieved state-of-the-art results on the leaderboard of the TUPAC16 Challenge for automatic methods (see Tab. 1), demonstrating the effectiveness of using multitask learning in combination with NIC to predict image-level labels from WSIs.

### 4.3. Image-level performance vs. number of tasks used in multitask training

The goal of the following experiment was to study the relationship between the number of tasks used to train the multitask encoder and the performance of the CNN regressor trained at image level, i.e. using TUPAC16 WSIs. First, we trained a set of encoders varying the number of tasks included during multitask training: 4 encoders using 1 task (only lymph, only mitosis, etc.), 6 encoders using 2 tasks (lymph+mitosis, lymph+prostate, etc.), 4 encoders using 3 tasks (lymph+mitosis+prostate, lymph+mitosis+colorectal, etc.), and 1 encoder using 4 tasks. Second, we compressed the TUPAC16 training dataset using each of the 15 previously trained encoders, and trained CNN regressors on them using 4-fold cross-validation as before in order to obtain an unbiased prediction for each training sample. Due to the large computational resources required to perform these steps, we used a reduced code size of $C = 16$.

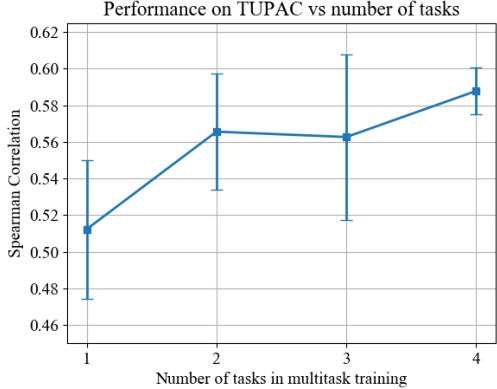 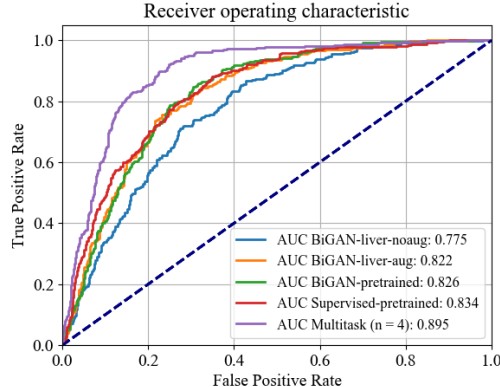

Figure 3: Relationship between the number of tasks used to train the multitask encoder and the performance of the CNN regressor trained on TUPAC16 (mean and std Spearman corr).

Figure 4: Predicting the presence of desmoplastic HGP in colorectal liver metastasis WSIs. Different encoding strategies are compared using the area under the ROC.

Table 2: Spearman correlation between task inclusion during multitask training, and performance at image level in TUPAC16

|  | Lymph | Mitosis | Prostate | Colorectal |
|---|---|---|---|---|
| **Correlation** | 0.319 | 0.033 | 0.077 | 0.824 |

We measured the Spearman correlation between the predictions of our system and the image-level labels, and averaged the results by the number of tasks (see Fig. 3). Note that we repeated the 4-task experiment four times with random weight initialization to obtain a more robust performance estimate. All the performance metrics are summarized in the Appendix B. We observed that increasing the number of tasks during multitask training produced a higher and more robust performance at image level. However, the large variance obtained in some cases (2 and 3 tasks) suggests that task selection might play an important role in the performance of multitask NIC.

Additionally, we measured the Spearman correlation between a binary variable describing whether a task was included during multitask training or not, and the performance of the system at image level. The results of this analysis are presented in Tab. 2. We found a positive correlation between the inclusion of the colorectal task and the global performance at image level. This result suggests that this dataset might be more valuable for feature extraction purposes than the rest. We recognize this task to be the most complex of all, requiring the encoder to extract robust features to accurately solve the classification problem. We hypothesize that multitask training can benefit from the highly specific features required to solve difficult classification tasks like this one.

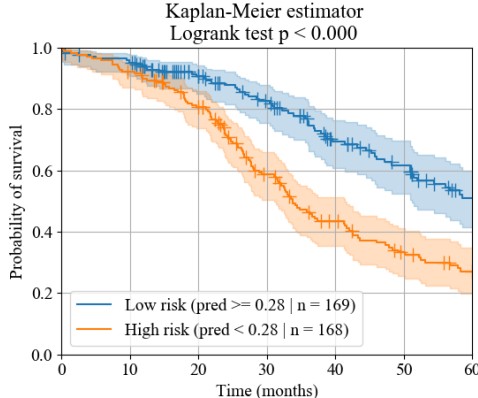

Figure 5: Predicting patient risk of death in colorectal liver metastasis WSIs. Learning from annotated HGP status.

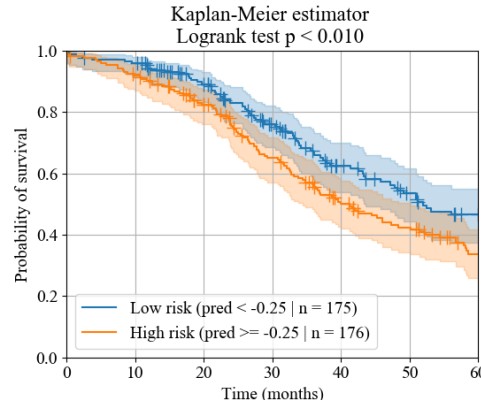

Figure 6: Predicting patient risk of death in colorectal liver metastasis WSIs. Learning from overall survival records.

### 4.4. Predicting patient risk of death in colorectal liver metastasis

**Desmoplastic HGP manual annotations**. We compressed all the liver WSIs using the multitask encoder introduced before in Sec. 4.1, and trained a CNN classifier to distinguish between dHGP or non-dHGP type on the compressed WSIs using 4-fold cross-validation (2 folds for training, 1 for validation, and 1 for testing). We measured the area under the ROC (AUC) on all the test samples to quantify performance.

In addition, we considered the predicted probability of dHGP as a proxy for the patient risk of death, and used the Kaplan-Meier (KM) estimator to model survival curves for two groups of patients, low and high risk, divided by the median predicted risk score.

For the dHGP classification task, we obtained an AUC of 0.895. Regarding the prognostic power of these predictions, results in Fig. 5 showed that our system could divide the population into two risk categories with high significance ($p < 0.001$).

**Overall survival records**. We trained a CNN classifier on the same compressed liver WSIs to predict patient risk of death learning directly from overall survival data (loss described in Sec. 3 under *Learning from patient overall survival*). As before, we used the KM estimator to assess the prognostic power of the classifier's predictions.

We found that this model was able to learn directly from overall survival data, dividing the population into two risk categories ($p < 0.01$), see Fig. 6. Note that no manual annotation was required on the colorectal liver metastasis dataset to perform this experiment, only patient records.

### 4.5. Comparing unsupervised and supervised encoders

We repeated the experiment described in Sec. 4.4 under *Desmoplastic HGP manual annotations* using different encoding options. In particular, we compressed the liver WSIs using several unsupervised and supervised encoding methods, and subsequently trained a CNN classifier to distinguish between dHGP and non-dHGP status.

We selected several encoders from the original NIC publication (Tellez et al., 2019) as baselines for the comparison: the unsupervised bidirectional generative adversarial network (*BiGAN-pretrained*) and the supervised network trained for lymph node tumor classification (*Supervised-pretrained*). Additionally, we trained two BiGAN encoders using patches extracted from the liver WSIs; in one case applying no augmentation during training (*BiGAN-liver-noaug*), and using heavy color augmentation in the other one (*BiGAN-liver-aug*). Finally, we also compared our proposed 4-task multitask encoder (*Multitask n = 4*).

Evidence in Fig. 4 highlighted three main results. First, heavy color augmentation played an important role in improving the features extracted by the encoder. Second, there seemed to be no difference between unsupervised and one-task supervised methods, trained on liver or any other organ. Three, multitask training substantially improved the overall performance of the system, obtaining the best classification AUC score of all tested methods (0.895).

## 5. Discussion

In this study, we extended Neural Image Compression (Tellez et al., 2019) by training the encoder with a supervised multitask learning approach. We trained the encoder to solve four classification tasks in Computational Pathology simultaneously, and used this model to perform the gigapixel image compression. First, supervised multitask training was key to obtaining a high performance at image level, surpassing unsupervised techniques. We found that increasing the number of tasks used to train the encoder was directly proportional to the system performance. Second, we obtained state-of-the-art results in predicting both the speed of tumor proliferation in invasive breast cancer (TUPAC16 Challenge), and HGP status in colorectal liver metastasis classification. These results in real-world tasks showcased the flexibility of multitask NIC as a method to empower WSI classification. Third, we used the proposed system to assess patient risk of death by learning directly from overall survival data, i.e. without human intervention. By doing so, we enabled the CNN classifier to work as an effective biomarker discovery tool for liver metastasis, moving beyond human knowledge rather than mimicking pathologists.

We acknowledge the main limitation of the proposed method to be a lack of straightforward criteria on how to expand the number and type of tasks used during multitask training, i.e. which tasks to select and include in the multitask loss function. We selected four representative tasks performed in the clinic with high-quality patch-level annotations. However, our results suggest that the WSI classifier might be sensitive to this choice. Careful weighting of multitask objectives and optimizing which tasks should be learned together is a matter of study in recent publications in the field (Kendall et al., 2018; Chen et al., 2018; Zamir et al., 2018). Future work should focus on conducting a more detailed evaluation on how to select these patch-level tasks, combining multiple objectives optimally, and including unsupervised or weakly-annotated data in the multitask loss.

## Acknowledgments

This study was supported by a Junior Researcher grant from the Radboud Institute of Health Sciences (RIHS), Nijmegen, The Netherlands; a grant from the Dutch Cancer Society

(KUN 2015-7970); and another grant from the Dutch Cancer Society and the Alpe d'HuZes fund (KUN 2014-7032); this project has also been partially funded by the European Union's Horizon 2020 research and innovation programme under grant agreement No 825292. The authors would like to thank Dr. Mitko Veta for evaluating our predictions in the test set of the TUPAC16 Challenge (Veta et al., 2019) dataset; and the developers of Keras (Chollet et al., 2015), the open source tool that we used to run our deep learning experiments.

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

Table 3: Predicting tumor proliferation speed in TUPAC16 (Spearman corr.) depending on which task was included during multitask training

| Lymph | Mitosis | Prostate | Colorectal | Correlation |
|-------|---------|----------|------------|-------------|
| No    | No      | No       | Yes        | 0.563       |
| No    | No      | Yes      | No         | 0.515       |
| No    | Yes     | No       | No         | 0.473       |
| Yes   | No      | No       | No         | 0.498       |
| No    | No      | Yes      | Yes        | 0.584       |
| No    | Yes     | No       | Yes        | 0.573       |
| No    | Yes     | Yes      | No         | 0.557       |
| Yes   | No      | No       | Yes        | 0.613       |
| Yes   | No      | Yes      | No         | 0.548       |
| Yes   | Yes     | No       | No         | 0.520       |
| No    | Yes     | Yes      | Yes        | 0.549       |
| Yes   | No      | Yes      | Yes        | 0.592       |
| Yes   | Yes     | No       | Yes        | 0.605       |
| Yes   | Yes     | Yes      | No         | 0.505       |
| Yes   | Yes     | Yes      | Yes        | 0.569       |
| Yes   | Yes     | Yes      | Yes        | 0.594       |
| Yes   | Yes     | Yes      | Yes        | 0.597       |
| Yes   | Yes     | Yes      | Yes        | 0.591       |

## Appendix A. Architecture and training details of the image-level CNN

The complete CNN architecture consisted of 8 convolutional layers using strided depthwise separable convolutions with 128 $3 \times 3$ filters, batch normalization (BN), leaky-ReLU activation (LRA), L2 regularization with $1 \times 10^{-5}$ coefficient, feature-wise 20% dropout, and stride of 2 except for the 7-th and 8-th layers with no stride; followed by a dense layer with 128 units, BN and LRA; and a final layer that depended on the application: a softmax dense layer for classification problems, and a linear output unit for regression tasks.

We trained the CNN using stochastic gradient descent with Adam optimization and 16-sample mini-batch, decreasing the learning rate by a factor of 10 starting from $1 \times 10^{-2}$ every time the validation metric plateaued until $1 \times 10^{-5}$. We minimized mean squared error for regression (TUPAC16), cross-entropy for classification (dHGP vs non-dHGP), and partial log-likelihood for patient risk prediction (targeting overall survival).

## Appendix B. Multitask training experiments

In Sec. 4.3, we experimented varying the number of tasks included during multitask training. We trained a set of encoders that were subsequently used to solve the TUPAC16 task. In Tab. 3, we show the performance obtained with each encoder.

## Appendix C. Learning from overall survival data

Survival analysis constitutes a regression problem where a model is trained to predict a risk score for each patient that is proportional to their chances of experiencing a given event,

in this case death. Each patient's WSI is associated with a label composed of two items: a follow-up period $t$ indicating the number of months that the patient has been enrolled in the study; and a binary variable $e$ that describes the event of whether the patient actually died or not. If a patient is event-free until her last follow-up, i.e. did not die, that sample is considered to be *censored* since we do not know when the event could take place in the future. The following introduces a loss function that enables a CNN to regress the patient's risk of death by exploiting censored and uncensored data.

The Cox's proportional hazards model (Cox, 1972) is the most widely used method to find the relationship between censored data and its covariates $x$. It models the hazard function $h(t, x)$ (probability of death at a given time) as a product between a time-dependent baseline $h_0(t)$ that is common to all patients, and a term proportional to the covariates weighted by learned coefficients $\beta$:

$$h(t, x) = h_0(t) \exp(\beta x). \tag{1}$$

We can estimate the optimal coefficients $\hat{\beta}$ by maximizing the logarithm of the partial likelihood:

$$\hat{\beta} = \arg\max \log \prod_{i \in D} \frac{\exp \beta x_i}{\sum_{j \in R_i} \exp \beta x_j} = \arg\max \sum_{i \in D} \left( \beta x_i - \log \sum_{j \in R_i} \exp \beta x_j \right), \tag{2}$$

where $D$ corresponds to the set of patients whose death is recorded (uncensored), and $R_i$ is the set of patients that did not experienced the event before patient $i$, i.e. survived longer than patient $i$.

In this study, the covariate vector $x$ is a compressed WSI $\gamma$ representing a patient. Instead of weighting each pixel with $\beta$ coefficients to produce a risk score, we parameterize this transformation with a CNN as $f(\theta, \gamma)$, with $\theta$ representing the trainable parameters of the neural network (Faraggi and Simon, 1995):

$$\hat{\theta} = \arg\max \sum_{i \in D} \left( f(\theta, \gamma_i) - \log \sum_{j \in R_i} \exp f(\theta, \gamma_j) \right). \tag{3}$$

Intuitively, by maximizing the previous term we enforce the CNN to learn a certain solution $\hat{\theta}$ that assigns high risk scores $f(\theta, \gamma)$ to those patients $i \in D$ that died early in comparison to those patients in each $i$'s risk set $R_i$ that survived longer, and thus deserve a lower risk score.

