# OpenReview forum: "Extending Unsupervised Neural Image Compression With Supervised Multitask Learning"
_MIDL.io/2020/Conference — MIDL 2020_

### Official Review · AnonReviewer2 · 2020-03-10
**the aim of this paper was developing unsupervised Neural Image Compression with supervised learning in prediction of risk of death in colorectal and liver metastasis patients.**

**Rating:** 3
**Confidence:** 3
**Recommendation:** Oral

**Summary:**

In this work, the author trained image compression using the multitask NIC and evaluated the obtained representations in two histopathology datasets that target imagelevel labels. First, they modeled the speed of tumor growth in invasive breast cancer. Second, they predicted histopathological growth patterns and the overall risk of death in patients with colorectal metastasis in the liver.

**Strengths:**

Paper is easy to follow with clear motivation about the proposed method.
Method is well validation with two different types of data.
Results on the evaluated metric shows the usefulness of the proposed method.
Limitations of the work are clearly noted.

**Weaknesses:**

Implementation details are not clear and should make the paper easily reproducible if the dataset is made publicly available.
The method should be expanded more to explain different experiments that have been used in this study.
The discussion part is very short without any rational that why their method can predict risk of death in colorectal and liver metastasis patients.

**Justification Of Rating:**

the authors accomplished their results and target associated with the particular goal being rated.
results met all standards, expectations, and objectives.
For overall performance, expectations were consistently met and the quality of work overall was good.

**Paper Type:**

both

**Questions To Address In The Rebuttal:**

I have one serious question. Figure 6 shows that logrank test p-value is less than 0.01 (significant difference between low and high risk group) while the confidential interval is CI: [0.99, 1.314]. How is it possible? If the CI cross 1, the p-value should be greater than 0.05. Please explain why the p-value is less than 0.01?

**Special Issue:**

no

---

> ### Author Response · Authors · 2020-03-27
> **Thank you for the questions**
>
> In relation to the weaknesses mentioned by the reviewer:
> Given the length limitation of 8 pages, we wrote the manuscript focusing most of the implementation details on novel contributions such as the multitask learning strategy. However, the rest of the image compression pipeline is described with great detail in a previous publication referenced in the paper (https://dx.doi.org/10.1109/TPAMI.2019.2936841). We implement the same image compression method as the one described in the original publication but use a different encoder network. If there is any specific question or confusion regarding the proposed methodology, please do not hesitate to submit a comment below. Regarding the discussion, we have expanded the text to account for the issues mentioned by this and other reviewers. Please note that the appendix contains plenty of details concerning how to model risk of death in the context of histopathology image analysis with deep learning.
>
> Questions:
> 1. I have one serious question. Figure 6 shows that logrank test p-value is less than 0.01 (significant difference between low and high risk group) while the confidential interval is CI: [0.99, 1.314]. How is it possible? If the CI cross 1, the p-value should be greater than 0.05. Please explain why the p-value is less than 0.01?
> The reviewer is correct in this analysis. However, please note that the p-value and the CI correspond to two different models (respectively), thus, they are independent from each other (in principle). First, we trained the multitask encoder in order to be able to compress whole-slide images (WSIs). Second, we compressed all the liver WSIs and trained a CNN on top of them targeting the patients’ risk of death. The output of this network is a set of predictions consisting of real-valued numbers that represent the probability of death of a given patient. Once we reach this point, we fit two independent models on these predictions (using overall survival as groundtruth data).
> The first model consists of the Kaplan-Meier (KM) estimator and the logrank test, the method that produces the survival curves. In order to fit this model, we need to divide the population into two groups, which we do by thresholding the predictions using the median value. The KM-logrank model tell us whether there is a significant difference between the two population groups in terms of prognosis (low and high risk), with a p-value for statistical significance. This is the p-value that the reviewer is referring to.
> The second model is the Cox proportional hazards (CPH). We use the continuous predicted score as a single covariate variable to fit the method. The output of this method is a hazard ratio and a confidence interval (CI), both reported in Figures 5 and 6 of the paper. This is the CI that the reviewer is referring to.
> In case of acceptance, we will improve the clarity of this part by removing the CPH model, which does not influence any of the conclusions of the paper. We would consider this analysis for future follow-up publications instead.

---

### Official Review · AnonReviewer1 · 2020-03-14
**Extending unsupervised NIC to multi-task supervised NIC improves performance on multiple downstream tasks**

**Rating:** 4
**Confidence:** 5
**Recommendation:** Oral

**Summary:**

In this paper, the authors aimed to improve the representations learned by Neural Image Compression (NIC) algorithms when applied to Whole Slide Images (WSI) for pathology analysis. The authors extended unsupervised NIC to a multi-task supervised system. A hard-parameters sharing network was presented with a shared, compressed representation branching out in task-specific networks. The authors evaluated the quality of these representations on multiple tasks, illustrating the added benefit of their multi-task system and the utility of using multiple tasks to supervised the feature extraction.

**Strengths:**

* This is a very well written paper. The introduction and description of the state of the art, in addition to the main limitations of popular algorithms is very clear and interesting to read. The experiments are clearly explained and the results are well presented.

* The decision to supervised the feature extraction in a multi-task setting is good and makes sense. Multi-task learning can extract a shared representation that is generalisable and this is evidenced in the results in the TUPAC16 set.

* Good and convincing results when compared to competing methods

* Strong validation

**Weaknesses:**

* It is a shame that the Kaplan-Meier estimator was not repeated for all baselines to further illustrate the strength of the multi-task features

* There are many more TUPAC16 results [http://tupac.tue-image.nl/node/62. https://arxiv.org/pdf/1807.08284.pdf] yet the presented method is benchmarked only against 3. It would be helpful to put the results in context with all other methods such as automatic and semi-automatic methods. Moreover, is there is a reason you did not validate on all TUPAC16 tasks?

**Detailed Comments:**

* Some interesting points are raised in the discussion about the number of tasks needed and which tasks are preferred. There have recently been papers focusing on determining which tasks should be learned together [1] and determining which tasks might benefit others [2]. It might be worthy to mention how these methods might be exploited in the future to determine how to pick the tasks for feature extraction to improve the WSI classifier.


[1] https://arxiv.org/pdf/1905.07553.pdf
[2] http://taskonomy.stanford.edu/

**Justification Of Rating:**

The is well written paper with a clear description of the state of the art and the reasoning behind the presented method. The method is well explained and the validation is strong with convincing results versus state of the art methods. The work also raises some interesting points regarding multi-task training for pathology and with further work could be a good paper.

**Paper Type:**

both

**Questions To Address In The Rebuttal:**

* Would it have been possible to also a decoder to the shared representation and learn feature reconstruction as an added loss to the multi-task objective?

* Given multi-task objective functions can be difficult to optimise, did you consider using methods that seek to dynamically weight each individual loss [1] or gradients of the loss [2]? It would be interesting to see if this improved the quality of the representations.

[1] https://arxiv.org/abs/1705.07115
[2] https://arxiv.org/abs/1711.02257

**Special Issue:**

yes

---

> ### Author Response · Authors · 2020-03-27
> **Thank you for the questions**
>
> 1. It is a shame that the Kaplan-Meier estimator was not repeated for all baselines to further illustrate the strength of the multi-task features
> We agree with the reviewer that this would be an interesting experiment to further characterize the impact of multitask learning in the method’s performance at image-level. However, we decided to postpone this exhaustive analysis and consider it for a follow-up publication. Attending to the mediocre performance obtained by the non-multitask methods in the other applications (see AUCs in Fig. 4), we argue that image compression without the multitask encoder would produce suboptimal results.
>
> 2. There are many more TUPAC16 results [http://tupac.tue-image.nl/node/62. (http://tupac.tue-image.nl/node/62.) https://arxiv.org/pdf/1807.08284.pdf] (https://arxiv.org/pdf/1807.08284.pdf]) yet the presented method is benchmarked only against 3. It would be helpful to put the results in context with all other methods such as automatic and semi-automatic methods.
> We have updated the manuscript to include more leaderboard entries in the manuscript table. Note that all the new methods produce worse results than the proposal except for the semi-automatic one (where regions of interest were selected by an expert pathologist).
>
> 3. Moreover, is there is a reason you did not validate on all TUPAC16 tasks?
> The unique feature of the proposed method is that we can train a model to make predictions of image-level labels even if these labels have unknown visual cues. Therefore, we selected the TUPAC16 proliferation score prediction task as a perfect fit for this use case. These labels are generated from a process involving molecular analysis, oblivious to any visual feature that might have been exploited by humans. We showed that our method can work as a knowledge discovery tool, identifying visual evidence for the task under consideration.
> In addition, please note that other tasks in TUPAC16 rely on mitosis detection, which is one of the tasks used during multitask learning. We avoided those tasks so that target image-level labels were fully independent from the patch-level annotations used during the multitask step.
>
> 4. Would it have been possible to also a decoder to the shared representation and learn feature reconstruction as an added loss to the multi-task objective?
> Yes, it is possible to add a reconstruction loss to the multitask objective (and any other unsupervised loss in general). As we have admitted before in this rebuttal and in the Discussion section, currently there is no data-driven criteria to perform an optimal task selection (loss selection).
> Initially, we hypothesized that adding an unsupervised loss term would not add extra specificity to the extracted features due to the lack of supervision, i.e., would not improve the performance of the supervised tasks. Thus, we decided to refrain from adding any reconstruction loss to the multitask objective and focus on supervised losses exclusively. We recognize, however, that this may be a valuable idea for future experiments, with the goal of characterizing the connection between the method’s performance and task selection.
>
> 5. Given multi-task objective functions can be difficult to optimise, did you consider using methods that seek to dynamically weight each individual loss [1] or gradients of the loss [2]? It would be interesting to see if this improved the quality of the representations. [1] https://arxiv.org/abs/1705.07115 [2] https://arxiv.org/abs/1711.02257
> This would definitely be an interesting extension to our paper, where we attempt to characterize the optimal strategy to perform multitask learning. However, our focus in this conference paper is to demonstrate that even the most basic multitask training setup outperforms all the previous unsupervisedly trained methods for gigapixel image compression.
>
> 6. Some interesting points are raised in the discussion about the number of tasks needed and which tasks are preferred. There have recently been papers focusing on determining which tasks should be learned together [1] and determining which tasks might benefit others [2]. It might be worthy to mention how these methods might be exploited in the future to determine how to pick the tasks for feature extraction to improve the WSI classifier. [1] https://arxiv.org/pdf/1905.07553.pdf [2] http://taskonomy.stanford.edu/
> We have extended the discussion stating promising lines of research, including the one mentioned by the reviewer in the question.

---

> > ### Comment · AnonReviewer1 · 2020-04-03
> > **Thank you for your comments**
> >
> > Thank you to the authors for their detailed comments. I am satisfied with their answers and look forward to seeing the extended Discussion section. I stand by my initial decision (Strong Accept).

---

### Official Review · AnonReviewer4 · 2020-03-14
**Interesting supervised approach to address neural image compression**

**Rating:** 3
**Confidence:** 4
**Recommendation:** Oral

**Summary:**

Histopathology image analysis is difficult because histology images usually are large and there is a substantial "noise-to-signal" ratio. As the authors mentioned, there is a need of efficient sampling methods to reduce the dimensionality of these images. The paper main hypothesis is that multitask supervised learning might allow to learn general and meaningful features.



**Strengths:**

- The paper address a complex and increasingly relevant topic in histopathology image analysis.

- The final supervised tasks are clinically relevant and results using the compression technique showed better performance in those final supervised tasks than the original NIC formulation or other approaches without compression.


**Weaknesses:**

- An analysis of which of the used supervised task in MTL are contributing the most in the final representation is missing.

- More information about the models performance on each of the different supervised tasks is needed.





**Detailed Comments:**

The authors propose to extend the neural image compression framework (NIC) by using supervised multitask learning task (MTL) in histology image analysis. The conclussions of the authors seem contradictory because they state that the representations learned by the MTL are at the same time (1) highly specific and (2) transferable. I would expect that a highly specific set of features  would not be able to transfer that well to another tasks. The authors do not provide a possible explanation to this seemingly contradictory result.

- Information regarding the models performance on each of the different supervised tasks is presented using only the accuracy metric. I think this metric might be not the most accurate to use in some of the tasks (i.e mitosis detection), in which there is a extreme unbalance in the classes. More information (such as F-measure) might be helpful to understand how well.

- Mitosis detection is a high resolution problem (usually performed at 40X) while prostate epithelium detection is done at a lower resolution (10X). The image patches come from different resolutions and from different tissues. The paper states that the image patches size is 64x64 at 0.5um/pixel. However, image patches for the mitosis task and for the prostate tissue in Figure 2. seem to have different magnification.

**Justification Of Rating:**

This is a really insightful paper demonstrating the ability of MTL framework in obtaining meaningful and general features for histological image compression. However, the paper contribution would be greater if additional details and evaluation regarding the different tasks used on the MTL framework are presented.

**Paper Type:**

both

**Questions To Address In The Rebuttal:**

- Which of the models trained on one supervised task obtain the best performance in the different final supervised tasks?

- What is the F-score of the mitosis detection task head?

- What is the correlation between the CNN regressor and each of the different compression models trained with different supervised task configurations?

**Special Issue:**

no

---

> ### Author Response · Authors · 2020-03-27
> **Thank you for your questions**
>
> 1. Which of the models trained on [...]
> We trained 4 encoders using each of the supervised tasks individually, and then used these networks as feature extractors for image-level regression to solve TUPAC16. Even though there are differences between the measured performances, we refrain from extracting conclusions from this observation. These differences might be explained by factors that are independent from the data distributions, e.g., different weight initializations between training runs. We acknowledge the need for a more systematic evaluation of the effects of task selection in the method’s performance, and we will consider it for a future extension of this paper.
> For your information, the Spearman correlation coefficients for each of these 4 systems are: lymph: 0.498, mitosis: 0.473, prostate: 0.515, and colorectal: 0.563. Additionally, we have updated the manuscript to include a table containing the performance of all the tested multitask configurations for TUPAC16.
>
> 2. What is the F-score of the mitosis [...]
> Since the patches have been extracted in a balanced manner (50% for each label), the F1-score is also high, similar to the accuracy: 0.9478. However, this score would look very different if we would test in patches sampled randomly (very imbalanced). Please note that evaluating the performance of alternative configurations of patch-level classifiers is out of the scope of this paper.
>
> 3. What is the correlation between the CNN [...]
> We trained and tested a total of 18 encoders, and updated the manuscript with a table containing the performance score for all of them. At the request of the reviewer, we have studied the correlation (Spearman) between the TUPAC16 performance metric and a binary variable indicating whether a given dataset was used during multitask training. In summary: lymph (0.319), mitosis (0.033), prostate (0.077), and colorectal (0.824).
> Remarkably, we found a positive correlation between the inclusion of the colorectal task and the global performance at image level (0.824). These results indicate that this dataset might be more valuable for feature extraction purposes than the rest. We recognize this task to be the most complex of all, requiring the encoder to extract robust features to accurately solve the classification problem. We hypothesize that multitask training can benefit from the highly specific features required to solve difficult classification tasks.  We have updated the manuscript to reflect this finding.
>
> 4. The authors propose to extend the neural [...]
> We define “specific” features as those that enable the downstream classifier to obtain a high classification performance; and “transferable” features as those that are useful for more than one task. When training a classifier, features are optimized to be specific, unless the loss is modified to account for other aspects, e.g., including a regularization term. Multitask training allows to optimize for both aspects directly, since features must be specific for all tasks, but also transferable in order to minimize the loss of the multiple classifier heads simultaneously.
>
> 5. Information regarding the models [...]
> We agree with the reviewer that other metrics rather than accuracy might be more descriptive for some of the tasks. However, this paper is not intended to be an analysis on individual task selection for multitask training. For this reason and to avoid unnecessary complexity, we sampled a balanced distribution of patches and decided to use accuracy as a common metric to report performance across all tasks. We believe that characterizing the relation between individual tasks and the system’s overall image-level performance is out of the scope of the current publication, and it is considered for future work.
>
> 6. Mitosis detection is a high resolution [...]
> The reviewer is correct in this appreciation. We decided to use 0.5 um/px (commonly known as 20X) as an intermediate resolution where both problems can be solved with an acceptable performance. However, the figure shows different magnifications due to an error with the image editing software. We have checked the resolution of the actual training patches and they are correct. We have fixed the figure to resemble the correct resolution as suggested by the comment.

---

### Official Review · AnonReviewer3 · 2020-03-14
**Interesting paper. Overall useful insights about the encoding spatial information. End to end system.**

**Rating:** 4
**Confidence:** 4
**Recommendation:** Oral

**Summary:**

The authors present an efficient procedure for doing Neural Image Compression. The idea is to tune the lower layer features by biasing them towards some downstream classification tasks. This retains useful discrimnative information while compressing the content. This has been observed across come vision tasks especially in scene parsing, and the paper provides application in medical imaging as well.

**Strengths:**

This is a well written paper. Clear explanations in terms of what is needed to address the WSI dimensionality issue. Evaluations seems to suggest the benefits of the proposal. End to end system. Has good impact.

**Weaknesses:**

From the perspective of the technical presentation and motivation there is not mot much weak aspects here:

What was the rationale for using these 4 specific tasks? Anything unique about them that would drive the NIC? In Figure 3, and section 4.3 what was the rationale for the jump from 3 to 4 tasks? The per task accuracies are very high, so I wonder if the tasks themselves are reasonably easy when trained independently i.e., the proposal is task sensitive.

What do you mean by not shared in section 3 first para? The tasks are trained together correct with the same encoded representation? So they are shared?

**Justification Of Rating:**

See above:
Specifically --- Clear explanations in terms of what is needed to address the WSI dimensionality issue. Evaluations seems to suggest the benefits of the proposal. End to end system. Has good impact.

**Paper Type:**

both

**Special Issue:**

no

---

> ### Author Response · Authors · 2020-03-27
> **Thank you for your questions**
>
> 1. What was the rationale for using these 4 specific tasks? Anything unique about them that would drive the NIC?
> As we mention at the end of the Discussion section, unfortunately we lack of data-driven reasons to select certain tasks among others. However, we have made our decision following two heuristic motivations.
> First, since high performance classifiers are usually obtained with highly-curated labels, we hypothesized that multitask training can also benefit from the same. Therefore, in order to build the best possible feature extractor, we selected the tasks with the highest-quality patch-level annotations that we could find, e.g., discarding weakly labeled data.
> Second, we were looking for representative tasks from Computational Pathology from multiple organs, so that the multitask mechanism could perceive the most tissue variability.
> All things considered, the four tasks presented in the paper satisfied our heuristics, however, we admit that a more thorough analysis on task selection could be an interesting extension to this paper.
>
> 2. In Figure 3, and section 4.3 what was the rationale for the jump from 3 to 4 tasks?
> The core idea of the paper is to prove that multitask training produces a NIC encoder that is superior to those trained unsupervisedly. We considered varying the number of training tasks as an interesting ablation study in order to understand how task selection could impact the system’s performance. The results indicate that the proposed method is sensitive to task selection and it is unknown why it performs better with 4 than 3 tasks. We hypothesize that more tasks lead to more specific and transferable features since the shared multitask network is enforced to extract features that must satisfy all downstream classifiers (removing features representing spurious patterns).
>
> 3. The per task accuracies are very high, so I wonder if the tasks themselves are reasonably easy when trained independently i.e., the proposal is task sensitive.
> We agree with the reviewer that the proposed method is sensitive to the tasks, including individual task performance. With the current results, we cannot derive any solid conclusion about the relation between patch-level classification accuracy and gigapixel-level performance. We acknowledge that further experimentation with these variables would be very interesting.
> Because of some questions from this and other reviewers, we have analyzed the correlation between including a certain task in the multitask objective and the resulting image-level performance. We found that including the colorectal task is positively correlated with the final performance of the system. This result suggests that extracting features related to tissue types is useful during histopathology image compression, therefore, more of these tissue classification tasks should be considered in the future.
>
> 4. What do you mean by not shared in section 3 first para? The tasks are trained together correct with the same encoded representation? So they are shared?
> The CNN used for multitask patch classification is divided into two components: the feature extractor and the classifier head. The first one is shared across all tasks, i.e., it is applied to patches from all datasets to extract features. The classifier heads are trained independently from each other (not shared), e.g., the mitosis head sees patches from the mitosis dataset only, and computes a mitosis classification loss accordingly. During backpropagation, the gradients of the loss w.r.t. the parameters of the heads are computed independently from each other (unaffected by other classifier heads). However, gradients w.r.t. the parameters of the shared feature extractor are computed and aggregated from the four loss signals. We have updated the manuscript to clarify this question.

---

### Author Response · Authors · 2020-03-27
**Thank you to reviewers**

Thank you to all the reviewers for the insightful questions and observations. Please find answers to all of the individual questions below. We are modifying the manuscript to reflect these answers accordingly. The updated version of the manuscript will be available if the paper is accepted.

---

### Meta-Review · Area_Chair1 · 2020-04-04
**MetaReview of Paper96 by AreaChair1**

**Rating:** 4
**Recommendation For Accepted Papers:** Oral

**Metareview:**

There is a high demand in histopathology image analysis to reduce or compress the (originally very large) image size. This paper deals with improving the representations learned by Neural Image Compression (NIC)  algorithms via multitask supervised learning. Results are provided on 2 datasets and 3 tasks.

The reviewers have collectively acknowledged the paper as well-written, addressing a hot topic in histopathology image analysis. The work is of high quality with excellent analysis of related works, soundness of the proposed approach, extensive and well-conducted validation, and finally conclusive results leading to a clear impact of the proposed method.

Weaknesses differ depending on the reviewer: they relate primarily to the clarity of the implementation details, rationale behind some choices (with regard to the tasks at hand), additional leaderboard entries, additional results or missing analysis.

The authors have replied precisely and made the required clarifications, and for most remarks, have updated the manuscript to clarify the questions. In some cases, they left the questions out their manuscript and carefully justified why.

Given both the original reviewers rating and their associated confidence level, as well as the authors rebuttal, I strongly recommend this paper to be accepted.

**Paper Type:**

both

**Special Issue:**

no

---

### Decision · Program_Chairs · 2020-04-11

Accept